# An Innovative Interactive Narrative Persona (INP) Approach for Virtual Reality-Based Dementia Tour Design (VDT) in Rehabilitation Contexts

**DOI:** 10.3390/bioengineering12090987

**Published:** 2025-09-17

**Authors:** Yuk Ming Tang, Suet Yi Tse, Hoi Sze Chan, Ho Tung Yip, Hei Tung Cheung, Mohammed Woyeso Geda

**Affiliations:** 1School of Management, Guangdong University of Science and Technology, Dongguan 523000, China; yukming.tang@polyu.edu.hk; 2Department of Industrial and Systems Engineering, The Hong Kong Polytechnic University, Hong Kong, China; hs-celia.chan@polyu.edu.hk (H.S.C.); hilary.cheung@connect.polyu.hk (H.T.C.); mohammed.geda@polyu.edu.hk (M.W.G.); 3The Jockey Club School of Public Health and Primary Care, The Chinese University of Hong Kong, Hong Kong, China; 1155187639@link.cuhk.edu.hk

**Keywords:** virtual dementia tour, interactive narrative persona, virtual reality, public awareness, empathy

## Abstract

The rising prevalence of dementia has raised significant public concern globally. However, the level of understanding and assistance concerning people with dementia remains limited. This study investigates the impact of virtual reality (VR) on enhancing public awareness and empathy toward dementia. We developed a Virtual Dementia Tour (VDT) designed to simulate the cognitive and sensory impairments associated with dementia while incorporating interactive decision-making elements. A total of 80 adults aged 18 years or older, residing in Hong Kong and with no personal or close family history of dementia or caregiving experience, were recruited for the study. Quantitative data were analyzed using paired and independent samples *t*-tests to assess the changes in the Dementia Attitudes Scale (DAS) scores before and after the intervention. The results indicate that the baseline awareness and understanding of dementia were low in both groups. However, participants who experienced the VDT showed significantly greater improvements in knowledge and empathy compared to the control group. The findings suggest that VR interventions can effectively promote dementia awareness, reduce stigma, and contribute to the development of dementia-friendly societies. This study contributes to the growing body of evidence supporting the use of VR as an innovative tool for advancing social awareness and empathy in public health education.

## 1. Introduction

Dementia is a neurodegenerative disorder characterized by progressive cognitive decline and impaired daily functioning [1]. It has emerged as a significant public health priority due to its escalating prevalence [2]. According to the World Health Organization (WHO), the number of individuals affected by dementia will reach 139 million by 2050 [3]. This trend highlights the urgent need to enhance public awareness, improve early detection, and strengthen support for people living with dementia and their caregivers. However, despite the growing recognition of dementia as a public health concern, public awareness and understanding of the condition remain inadequate [4]. Misconceptions, stigma, and negative stereotypes continue to hinder timely diagnosis, restrict access to appropriate care, and limit social inclusion for those affected [5].

In recent years, government and non-governmental organizations have stepped up initiatives aimed at raising awareness, reducing stigma, and promoting public discussion around dementia-related issues [6]. For instance, campaigns such as the UK’s “Dementia Friends” initiative, led by the Alzheimer’s Society, provide informational sessions and online resources to educate communities about dementia [7]. Similarly, media coverage is commonly used to raise awareness and contribute to reducing stigma, as well as to foster public discussion regarding dementia-related issues. Campaigns, such as Australia’s “Dementia Australia” advertisements, use television and social media to disseminate facts about dementia and encourage supportive behaviors [8,9]. Educational programs have also been integrated into school curricula and community centers, such as the “Dementia Awareness in Schools” program in Canada, which introduces students to basic concepts about dementia to foster empathy and reduce stigma among younger generations [10].

However, the effectiveness of these traditional approaches has been mixed, and significant gaps in public understanding persist. For instance, evaluations of public awareness campaigns, such as the “Dementia Friends” initiative, have shown modest improvements in knowledge but limited impact on reducing deep-seated stigma toward people with dementia [11]. Similarly, media campaigns often struggle to reach diverse populations, with studies indicating that stigma associated with dementia, particularly within culturally diverse communities, remains unaddressed. This includes beliefs in some communities that dementia is a normal part of aging or a result of personal failure [12]. Moreover, traditional approaches such as leaflets frequently fail to convey the lived experiences of people with dementia, limiting their ability to foster empathy or challenge stereotypes effectively [13]. These limitations underscore the need for innovative, experiential interventions to bridge the gap between factual knowledge and emotional understanding of dementia.

Virtual reality (VR) technology has emerged as a promising tool for immersive and experiential learning across various domains, including healthcare and education [4,14,15]. Several studies have explored the application of VR in dementia education and awareness, highlighting the potential of VR as an effective tool for enhancing public understanding and empathy toward dementia [16,17,18]. By allowing users to virtually experience the daily challenges, cognitive difficulties, and emotional realities faced by those with dementia, VR can overcome many limitations of traditional educational approaches [18]. Despite these advantages, most existing VR methods in dementia education focus primarily on linear scenario-based simulations [19] or clinical training for healthcare professionals [18,20,21,22], missing the dynamic narrative structures of dementia experiences. Furthermore, VR-based interventions emphasize general empathy-building and environmental challenges [23], with limited personalization to dementia stages [24]. This limitation can reduce the authenticity and relatability of the simulated experiences.

To address the limitations of current VR interventions in dementia education, recent research emphasizes the importance of developing evidence-based personas that capture the varying needs and experiences of people with dementia, ensuring that the technology design is both authentic and person-centered [25,26]. Incorporating personas into technology design helps to ensure more authentic, relatable, and effective user experiences. In particular, those centered on everyday conversational “small stories” have been recognized as crucial for sustaining personhood, promoting citizenship, and fostering empathy in dementia care environments [25]. These approaches align with the principles of person-centered care by valuing both verbal and non-verbal expressions, and by constructing environments where individuals with dementia can actively participate in shaping their own narratives.

While interactive narrative frameworks have been explored in therapeutic contexts to stimulate memory and communication [27], their potential for public education has not been fully realized, particularly with regard to integrating user agency and decision-making. In response, this study introduces and evaluates an innovative VR-based intervention for the general public, utilizing an Interactive Narrative Persona (INP) framework within the Virtual Dementia Tour (VDT). The INP framework is designed to advance VR-based dementia education by uniquely combining the following:i.Evidence-based persona development to authentically represent individuals at different dementia stages and with varied symptom profiles.ii.Dynamic, branching narrative structures that allow users to make meaningful deci-sions and experience the evolving, context-dependent realities of living with demen-tia.iii.Personalized, first-person engagement, enabling users to better understand the emo-tional, social, and cognitive challenges faced by people with dementia.iv.Inclusion of both symptom experiences and coping strategies, offering a holistic per-spective that goes beyond traditional symptom simulation.

Compared to existing VR design methods, the proposed INP framework offers several distinct advantages:It supports mindful design by encouraging stakeholders to consider critical factors when creating environments for people with dementia, resulting in greater authenticity and relatability when using the INP rather than generic avatars [28].It improves social and relational engagement by fostering environments that support storytelling, memory stimulation, and communication [25,28].It allows users to participate actively in narrative experiences tailored to diverse dementia profiles [27].It offers broader applicability for the general public by addressing key gaps in community-based dementia awareness.

The remainder of this paper is organized as follows. Section 2 outlines the proposed INP approach for the design of the VDT. Section 3 describes the VDT training program implemented in this study, and present the results of the case study and discuss their implications, respectively. Section 4 addresses the study’s limitations and suggests directions for future research. Finally, Section 5 concludes with a summary of the main findings.

## 2. Interactive Narrative Persona (INP) Approach for VDT Design

### 2.1. Overview of the INP Framework

Conventional dementia awareness methods such as leaflets, public campaigns, and school programs primarily deliver factual information and often fail to foster genuine empathy [13]. In contrast, an Interactive Narrative Persona (INP) is a virtual character designed to embody the lived experiences, emotions, and challenges of real individuals [29]. Hence, by placing users in the role of a person living with dementia, INPs enable participants to make meaningful decisions and experience firsthand the emotional and cognitive challenges faced daily, fostering deeper empathy and understanding. The INP framework serves as the conceptual foundation for our Virtual Dementia Tour (VDT) design. As illustrated in Figure 1, the framework integrates three core components, persona creation, narrative structure, and interactive mechanics, to create an immersive and educational VR experience. The main functions and goals of the components are summarized in Table 1.

The VDT utilizes a series of interactive simulations, each designed to realistically portray daily challenges encountered by people living with dementia from a first-person perspective [30]. In developing the VDT, we grounded each scenario in detailed user personas, ensuring representation across diverse backgrounds and disease stages. To develop these personas, we conducted qualitative research, including interviews and ethnographic observations, to capture authentic experiences and motivations.

We then incorporated dynamic narrative elements such as branching storylines and decision points throughout the VR environment. This design transforms users from passive recipients into active participants, enhancing engagement, agency, and personalization. By integrating persona-driven insights, emotionally rich storytelling, and interactive engagement, we created an INP framework that serves as a methodology for fostering empathy, understanding, and compassion toward individuals living with dementia.

### 2.2. Persona Creation

Persona creation forms the basis of the INP framework, ensuring that the VDT experience is grounded in the diversity of real-world dementia cases. The design of personas and scenarios in our VR-based VDT was grounded in Kitwood’s theory of person-centered care, a leading framework in dementia care. Kitwood’s concept of personhood emphasizes the importance of recognition, respect, and trust in supporting the well-being of people with dementia. This approach ensures that each persona’s narrative addressed core psychological needs identified by Kitwood: comfort, attachment, inclusion, occupation, and identity, which are critical for person-centered dementia care [31,32]. In addition to Kitwood’s theory of person-centered care, we drew on the biopsychosocial–ecological framework proposed by Podgorski et al. [33]. Their framework emphasizes the interplay of biological, psychological, and social factors—including family dynamics and caregiving networks—in shaping the lived experience of dementia. These two frameworks guided our persona creation and narrative design to ensure that medical, functional, emotional, and relational aspects of dementia were authentically represented.

To develop the personas, we employed a qualitative research approach. We conducted interviews and gathered data from individuals living with dementia, their caregivers, and healthcare professionals [22]. Through thematic analysis of survey responses, we identified key demographic and psychosocial attributes, including gender, age, occupational background, stage of dementia, behavioral patterns, and emotional states. These insights informed the construction of detailed persona profiles that reflect both genders and various stages of dementia, ensuring an inclusive and comprehensive depiction of individuals living with the condition. Examples of these personas are provided in Figure 2 and Figure 3. “Sarah Lee”, a 76-year-old retired principal at the moderate stage of dementia, illustrates the role of family support and the need for emotional and safety measures in daily life (see Figure 2). In contrast, “Thomas Kwok,” a 61-year-old accountant with mild dementia, highlights the importance of cognitive stimulation and memory strategies (see Figure 3). Names and photos have been altered to protect the privacy of the individuals depicted.

By featuring diverse personas, the VDT challenges gender bias and promotes a broader understanding of dementia as a condition not confined to any one gender, thereby fostering empathy and increasing awareness of the varied experiences of dementia patients. These personas guide the development of narrative branches and interactive touchpoints, enabling participants to step into the shoes of individuals at different stages of dementia and fostering empathy through firsthand virtual experiences [34].

We also included personas at different stages of dementia, specifically mild and moderate, to illustrate the progression of the disease and its distinctive impact on daily life [26]. This approach enables users to gain a comprehensive understanding of how symptoms and needs change as dementia advances, providing a realistic portrayal of the disease’s evolution. Through the inclusion of multiple personas with varied backgrounds and disease stages, user engagement and personalization can be enhanced. This diversity allows users to form stronger emotional connections based on their own experiences or those of loved ones, which in turn encourages deeper involvement with the simulation [35]. Moreover, presenting multiple storylines helps prevent monotony and maintains user interest throughout the VDT experience [36].

### 2.3. Narrative Design

Narrative design in the INP framework is a cornerstone of the VDT, focusing on the creation of cohesive, emotionally resonant, and immersive storylines that unfold within the VR environment. The narrative design ensures that users are not merely passive observers but active participants who shape the experience through their decisions [37]. This approach aims to deepen users’ understanding of dementia by simulating the lived experiences of individuals with the condition, fostering empathy, and challenging stereotypes.

The key features of the narrative design include character development, realistic scenarios, and branching dialogue trees, all of which drive emotional engagement in the user experience (Figure 4). Our narrative design brings each persona to life with a detailed backstory, demographic profile, and unique set of challenges. Each persona’s journey highlights different aspects of dementia. A central feature of our narrative design is the branching dialogue tree, which provides dynamic decision points throughout the experience. At each juncture, users select from multiple responses, and these choices directly shape the unfolding narrative. For example, when a persona with dementia is questioned about a misplaced item, the user might choose between “I don’t know where it is” or “Someone must have taken it.” Each option triggers a different non-player character (NPC) reaction, altering the emotional tone and direction of the scene.

### 2.4. Interactive Design

In our method, we center the VDT experience on eight interactive touchpoints. These touchpoints form the core of the interaction design, each carefully designed to immerse users in the everyday challenges faced by individuals with dementia [27]. We use these touchpoints to structure the interaction design, guiding participants through decision-making scenarios that reflect the lived realities of dementia.

At each touchpoint, we present users with context-specific prompts through Unity’s interactive buttons. We ensure that each user’s decision produces immediate effects within the simulation. For instance, when users are prompted to accept or reject social support, selecting an option triggers a script to perform the following tasks:Change the animation and dialogue of NPCs,Update the scenario’s narrative branch,Adjust the background music or sound effects to reflect the emotional tone.

This dynamic, user-driven approach ensures that participants move beyond passive observation and become active agents within the simulation. Their choices shape the flow and outcome of each session, personalizing the experience and encouraging meaningful engagement and empathy. Table 2 summarizes these touchpoints, the dementia symptoms they address, and their implementation in Unity.

### 2.5. Technical Development and Integration

The technical development of the VDT closely follows the INP framework, ensuring that the persona, narrative, and interaction elements are preserved throughout production. The process is organized into six distinct phases, from scenario recording to VR deployment, as illustrated in Figure 5 and summarized in Table 3.

In addition to the INP framework, the design of the VDT also draws upon the conventional 3I model of virtual reality systems, comprising Immersion, Interaction, and Imagination, as shown in Figure 6. Immersion was achieved through the use of 360-degree panoramic video, spatial audio, and high-fidelity environmental rendering, which collectively enabled users to feel fully embedded in the experiences of the personas. Technological immersion—covering aspects such as display resolution, field of regard, and multisensory consistency—has been recognized as essential for creating presence in VR systems [38].

Furthermore, interaction was implemented via user-driven decision points embedded in Unity, allowing participants to influence the narrative direction and character responses. Research shows that perceived interactivity in VR enhances learning outcomes and engagement [39,40]. Imagination was activated by placing users in emotionally resonant and cognitively complex scenarios, encouraging them to adopt the perspectives of individuals with dementia and reconstruct their inner experiences. Story-driven immersive environments combining imaginative stimuli and user decisions have been shown to cultivate empathy in nursing and dementia-related education [41].

By aligning the INP framework with the 3I model, we ensured that both the narrative and technical dimensions of the VR system synergistically enhanced the user experience. While the INP approach enabled empathetic engagement through persona-based storytelling and structured decision-making, the 3I model reinforced the cognitive and emotional impact of the simulation through sensory immersion and imaginative projection. Together, these frameworks provided a robust foundation for developing a transformative and impactful dementia education tool.

## 3. Experimental Design and Results

### 3.1. Study Design

We follow a structured experimental methodology to assess the effectiveness of our VDT intervention compared to traditional dementia awareness methods. The overall procedures are illustrated in Figure 7.

### 3.2. Participant Recruitment and Group Assignment

We recruited a total of 80 participants using convenience sampling and clear inclusion and exclusion criteria. Eligible participants were adults aged 18 or above, residents of Hong Kong, and with no personal history of dementia. To minimize bias or emotional distress, we excluded individuals who were current or past caregivers of people with dementia or had close family members with the condition. This approach was chosen to create a more homogeneous baseline and to avoid the confounding effects of prior experience. All participants provided informed consent prior to data collection. Random assignment to the experimental and control groups was conducted using a computer-generated random number sequence to ensure allocation concealment and minimize selection bias. Participants were divided as follows:Experimental group (*n* = 40): Participants experienced the Virtual Dementia Tour (VDT) simulation.Control group (*n* = 40): Participants read an informational leaflet about dementia, with content synchronized to the VR simulation to ensure consistency.

The control group was established to evaluate the effectiveness of traditional dementia awareness methods against the immersive VDT experience. By providing a leaflet with content matched to the VR scenario, we ensured that both groups received the same factual information, isolating the effect of the delivery format. This design strengthens the validity of our findings by attributing the observed differences in outcomes specifically to the VR intervention. The leaflet content is shown in Figure 8 and Figure 9.

### 3.3. Questionnaire Design

All participants completed the Dementia Attitudes Scale (DAS), both before and after their assigned intervention [42]. The DAS is a well-validated instrument that enables us to accomplish the following:Assess attitudes: Systematically evaluate public perceptions and beliefs about dementia.Identify knowledge gaps: Highlight misconceptions or stigma, informing targeted educational interventions.Monitor change: Allow direct pre–post comparison to measure the intervention’s impact.Provide standardized, quantitative data to inform future awareness campaigns or policy development.

The detailed survey items and scoring procedures for the DAS are documented in Section A.1. Additionally, a post-simulation questionnaire, outlined in Section A.2, was administered to evaluate participants’ perceptions of the VDT experience and its effectiveness in meeting the study objectives.

### 3.4. Results

#### 3.4.1. Descriptive Statistics

We analyzed all the collected data, following a comprehensive process that included data cleaning and preparation (e.g., handling missing values and outliers), descriptive statistics to summarize participant demographics and baseline attitudes, inferential statistics such as paired and independent samples tests to evaluate changes in the DAS scores within and between groups, and data visualization for clear presentation and interpretation of the results.

Table 4 presents the socio-demographic characteristics of the study sample. The majority were aged 18–25 (90%) and held a bachelor’s degree (76.3%), with a near-equal gender split (60% female, 40% male).

#### 3.4.2. Pre- and Post-Intervention Comparison (Within-Group Analysis)

The pre- and post-intervention DAS scores were compared within both the control group (leaflet intervention) and the experimental group (VDT intervention) to assess the effectiveness of each approach in shaping attitudes toward dementia. To ensure the validity of the paired samples *t*-tests, we first tested the assumption of sphericity for the paired differences using Mauchly’s test of sphericity [43], which was passed for both the control and experimental groups. We analyzed the changes in the DAS scores to evaluate the impact of each intervention. As shown in Table 5, the control group demonstrated an increase in the mean DAS score from 70.72 (SD = 13.49) before the intervention to 90.05 (SD = 12.95) after reading the leaflet. The experimental group exhibited a larger increase, from a mean pre-intervention DAS score of 74.75 (SD = 13.94) to a post-intervention score of 107.23 (SD = 10.85) following the VDT experience.

The paired samples *t*-tests (Table 6) confirmed that the post-intervention DAS scores were significantly higher than the pre-intervention scores in both groups (control: t = −11.53, *p* < 0.001; experimental: t = −15.61, *p* < 0.001). In both groups, the post-intervention DAS scores were found to be significantly higher than the pre-intervention scores, with improvements observed across all the DAS items following participation in either intervention (further illustrated in Figure 10).

#### 3.4.3. Comparison of Post DAS Scores Between Control and Experimental Groups

To evaluate the relative effectiveness of the two interventions, the post-intervention DAS scores were compared between the control group (leaflet intervention) and the experimental group (VDT intervention). This analysis aimed to determine which approach was more impactful in enhancing participants’ understanding and attitudes toward dementia. Prior to conducting an independent samples *t*-test, we tested the assumption of equal variances using Levene’s test [44]. As summarized in Table 5, the mean post-intervention DAS score was 90.05 (SD = 12.95) for the control group and 107.23 (SD = 10.85) for the experimental group. The independent samples *t*-test revealed a statistically significant difference between the groups (t = −6.35, *p* < 0.001), with the experimental group showing a greater improvement in attitudes. The mean change in the DAS score was substantially larger in the experimental group (+32.48) compared to the control group (+19.33), as shown in Table 7.

Correlation analyses (Table 8) further support the robustness of these findings, with significant associations observed between the pre- and post-intervention scores in both groups (control: r = 0.68, *p* < 0.001; experimental: r = 0.46, *p* = 0.003. These results confirm the superior effectiveness of the immersive VDT experience compared to the traditional leaflet approach in improving public attitudes toward dementia.

#### 3.4.4. Reliability and Validity

To ensure measurement consistency, reliability analyses were conducted for the DAS across all the items, both pre- and post-intervention. The internal consistency of the scale was found to be excellent, with Cronbach’s alpha values of 0.946 for pre-intervention and 0.957 for post-intervention administration. These results indicate high reliability for the DAS in both study phases.

Further item analyses demonstrated satisfactory corrected item–total correlations, confirming that each item contributed meaningfully to the overall scale. A radar chart (Figure 11) visualizes the corrected item–total correlations, revealing improved reliability post-intervention, though Items 16 and 17 show reduced contributions. The item statistics (means and standard deviations) support the scale’s robustness, affirming the VDT’s effectiveness in enhancing attitudes toward dementia.

## 4. Discussion

This study investigated the effectiveness of a VDT, utilizing an INP framework, in enhancing public awareness and empathy toward dementia. The findings demonstrate that both the VDT and conventional leaflet interventions led to significant improvements in participants’ attitudes. The results also emphasize the urgent need for more effective educational initiatives to improve public perception and foster a deeper understanding of the challenges faced by individuals living with dementia.

### 4.1. Theoretical Implications: Experiential Learning and Empathy Development

This study provides important theoretical contributions by demonstrating how an immersive INP framework can transform public attitudes toward dementia. Unlike conventional information-based approaches, the INP methodology is grounded in experiential learning theory, which emphasizes that people learn best through active participation and direct experience. Through immersion in realistic, first-person scenarios that capture the daily challenges of individuals with dementia, the INP framework enables deeper cognitive and emotional engagement than passive learning methods.

A central innovation of the INP approach is its use of diverse, rigorously developed personas and branching, decision-driven narratives. By incorporating diverse personas and branching narratives grounded in Kitwood’s theory of person-centered care and the biopsychosocial–ecological framework, our approach enables participants to experience the complex interplay of psychological, social, and biological factors that characterize dementia. This ensures that users are exposed to the complex and varied realities of dementia, promoting inclusive understanding.

Another key differentiator of the INP framework is its emphasis on interactive mechanics. Participants are not passive observers; instead, they are empowered to make meaningful choices at critical decision points, with each choice influencing the narrative and outcomes. This interactivity fosters agency and personalization, which are critical for empathy development. As users witness the consequences of their actions within the simulation, they are encouraged to reflect on the emotional and cognitive challenges faced by people with dementia.

Furthermore, by moving beyond scripted VR scenarios, the INP framework offers a transformative educational experience that is both immersive and personalized. It extends the literature on empathy cultivation and experiential learning by showing that dynamic narratives, authentic personas, and user-driven decision-making can create more impactful and lasting attitude change than traditional VR interventions.

### 4.2. Practical Implications for Dementia Awareness and Education

#### 4.2.1. Advancing Dementia Awareness and Education

Our findings demonstrate that the INP-based VDT is substantially more effective than traditional leaflet interventions in improving knowledge and fostering empathy. This is attributed to several unique features:Immersive and interactive experience: The VDT allowed participants to actively engage with realistic scenarios, making decisions that directly influenced outcomes. Such interactivity led to greater cognitive stimulation and deeper engagement compared to passive reading.First-person perspective: Experiencing dementia symptoms from a first-person viewpoint enabled participants to better appreciate the emotional and cognitive challenges faced by people with dementia. This perspective-taking is crucial for cultivating empathy and reducing stigma.Emotional connection: The VDT’s ability to evoke emotional responses through simulated real-life situations enhanced empathy, as reflected by over 70% of experimental group participants reporting increased understanding and compassion.Realistic representation: More than 80% of VDT participants agreed that the simulation accurately depicted the daily experiences and communication difficulties of people with dementia, underscoring the intervention’s authenticity and impact.

#### 4.2.2. Feasibility and Societal Impact

The wireless and user-friendly VR setup enhances accessibility, enabling implementation in community centers, schools, and even home environments. This supports broad integration into public health and educational initiatives. Moreover, the observed improvements in attitudes and empathy support the potential of VR interventions to promote dementia-friendly societies. By increasing public understanding and reducing stigma, immersive VR experiences can contribute to a more inclusive and supportive environment for individuals living with dementia. However, VR interventions should be integrated as part of comprehensive strategies involving policy, education, infrastructure, and social support.

#### 4.2.3. Relevance for Rehabilitation Practice

The INP-based VDT also holds significant potential for use in various rehabilitation contexts:Professional training: Enables health professionals and rehabilitation staff to better understand the lived experiences and psychosocial needs of people with dementia, enhancing empathy and person-centered care.Caregiver support: Assists family caregivers in anticipating behavioral and emotional changes, improving preparedness and coping strategies.Patient engagement: Elements such as scenario personalization and interactive decision-making can be adapted for cognitive rehabilitation and occupational therapy, providing engaging and meaningful activities for people living with dementia.The INP-based VDT framework can also serve as a shared platform to foster communication and understanding among healthcare providers, social workers, and families, supporting holistic approaches to dementia care and rehabilitation.

#### 4.2.4. Study Limitations

While the INP-based VDT demonstrated significant improvements in participants’ attitudes and empathy toward dementia, this study has some limitations:Sample characteristics: The study sample was predominantly young (90% aged 18–25) and well educated (76.3% with a bachelor’s degree), which may limit the generalizability of the findings.Short-term evaluation: The study relied on immediate pre- and post-intervention assessments. Long-term retention of attitude changes and behavioral impact were not evaluated.Technological barriers: While the VR setup was wireless and user-friendly, access to VR headsets may not be universal, potentially limiting the scalability in certain communities or resource-limited settings.

### 4.3. Recommendations and Future Directions

To strengthen the evidence base and practical applicability of VR interventions in dementia education, future research should recruit larger, more diverse, and more representative samples, including older adults and individuals from varied backgrounds. It is essential to include groups such as caregivers and healthcare professionals to more fully evaluate the effectiveness and relevance of the INP framework across different populations. They should also conduct longitudinal assessments to evaluate the persistence and sustainability of attitude changes over time. The scalability, cost-effectiveness, and integration of VDT interventions within broader community and educational frameworks should also be explored. Furthermore, future studies should explore collaboration mechanisms among healthcare professionals, policymakers, and community organizations to maximize the reach and effectiveness of VR-based dementia education.

## 5. Conclusions

This study demonstrates the transformative potential of the VDT, an innovative VR-based intervention utilizing the INP framework, in terms of enhancing public awareness and empathy toward dementia. Compared to traditional educational methods, the VDT provided a significantly greater improvement in participants’ knowledge, attitudes, and emotional understanding. The immersive design, featuring diverse personas, branching narratives, and interactive touchpoints, effectively bridged the gap between factual knowledge and emotional understanding, addressing the limitations of conventional awareness campaigns. These findings contribute to the growing evidence base for VR-based experiential learning tools to reduce stigma and foster dementia-friendly societies. Continued development and wider implementation of such interventions hold promise for advancing societal understanding and inclusivity for people living with dementia.

## Figures and Tables

**Figure 1 bioengineering-12-00987-f001:**
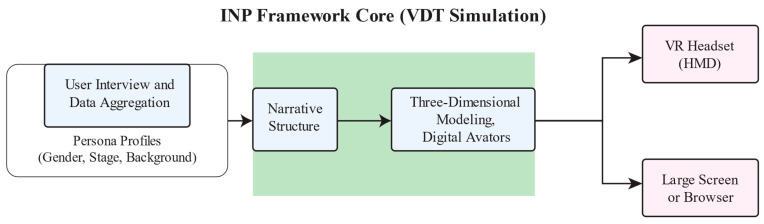
The framework of the proposed VDT design approach.

**Figure 2 bioengineering-12-00987-f002:**
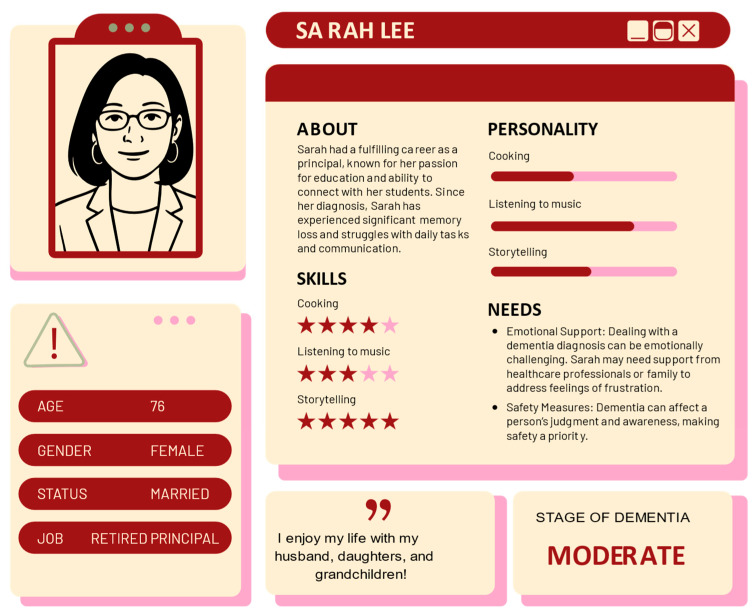
Persona profile: moderate stage dementia.

**Figure 3 bioengineering-12-00987-f003:**
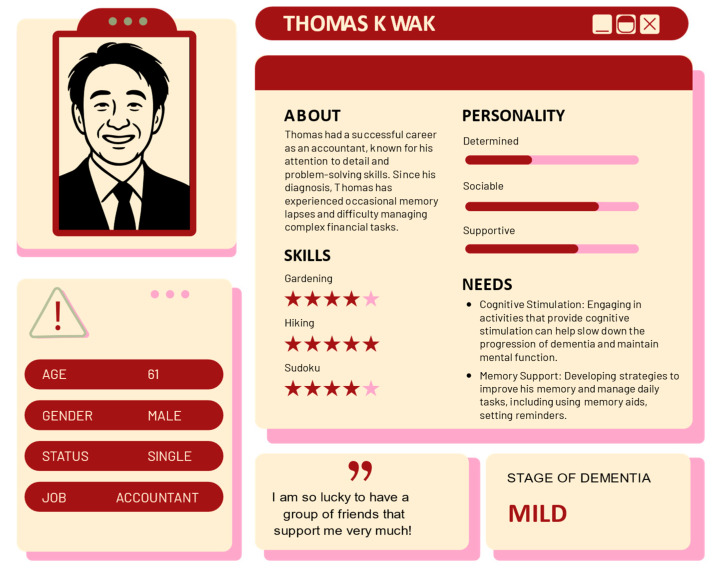
Persona profile: mild stage dementia.

**Figure 4 bioengineering-12-00987-f004:**
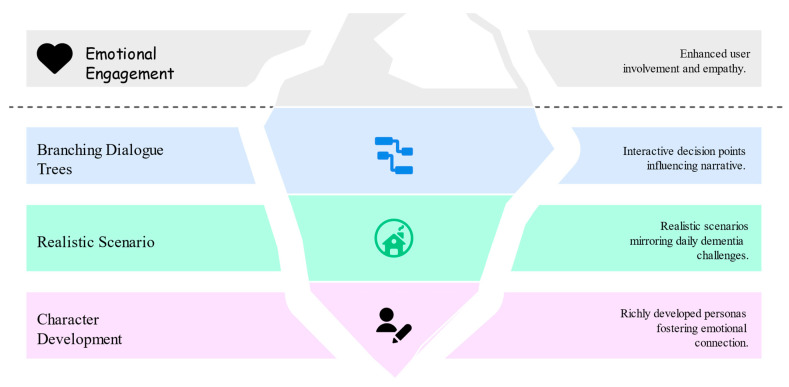
Concept map of the narrative design components in the INP framework.

**Figure 5 bioengineering-12-00987-f005:**
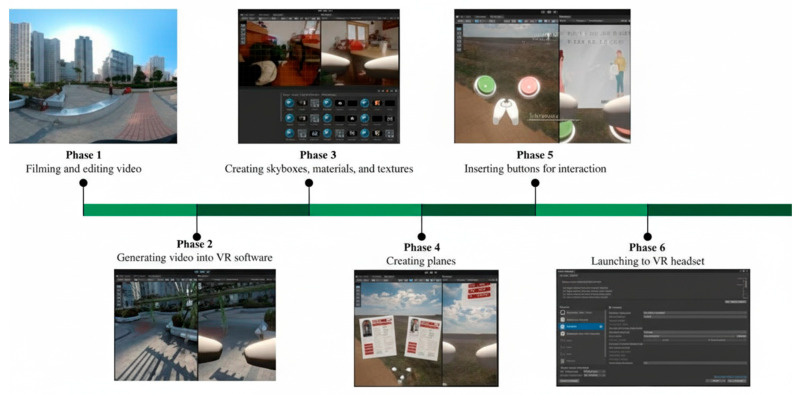
Phases of VDT development.

**Figure 6 bioengineering-12-00987-f006:**
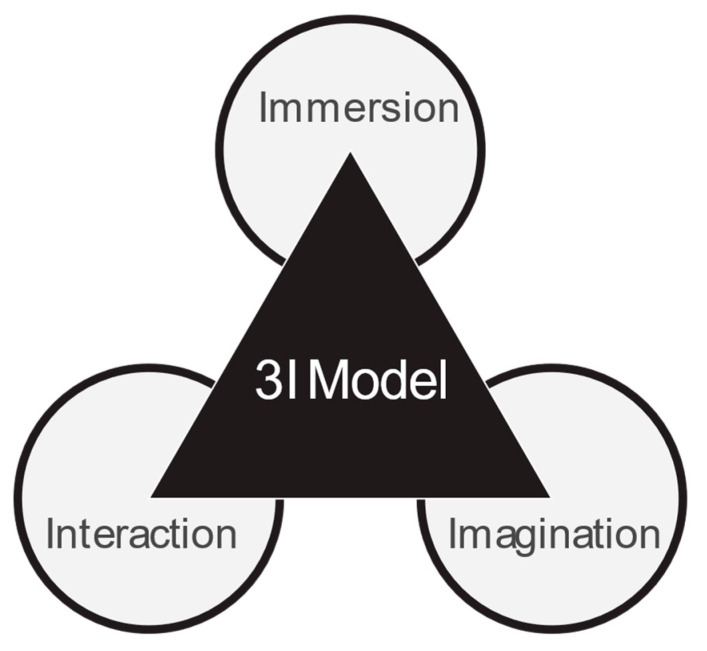
The 3I model of virtual reality.

**Figure 7 bioengineering-12-00987-f007:**
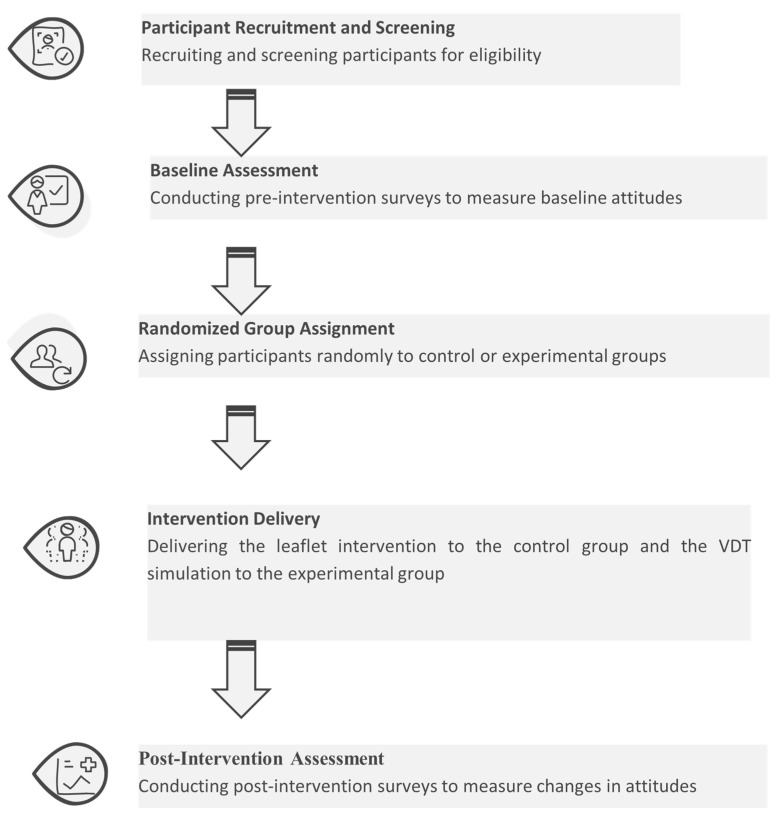
Flowchart of the study design.

**Figure 8 bioengineering-12-00987-f008:**
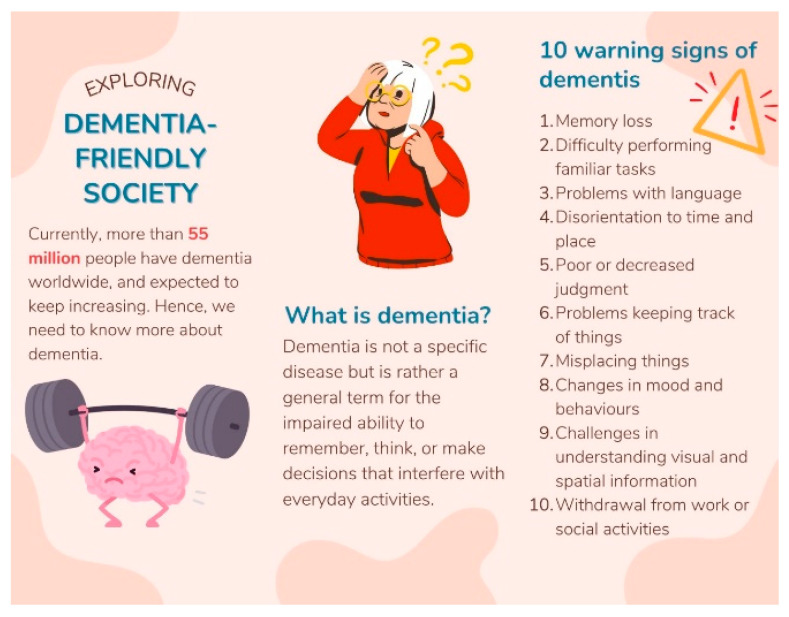
First page of the leaflet.

**Figure 9 bioengineering-12-00987-f009:**
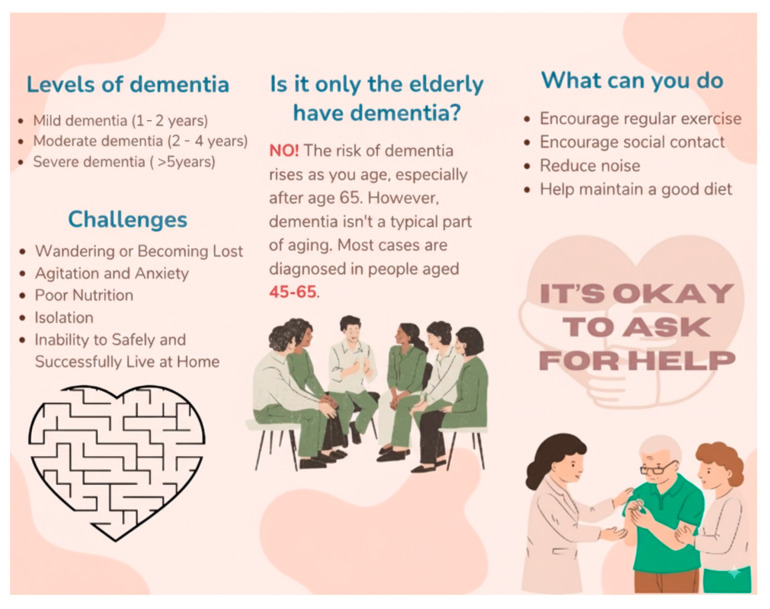
Second page of the leaflet.

**Figure 10 bioengineering-12-00987-f010:**
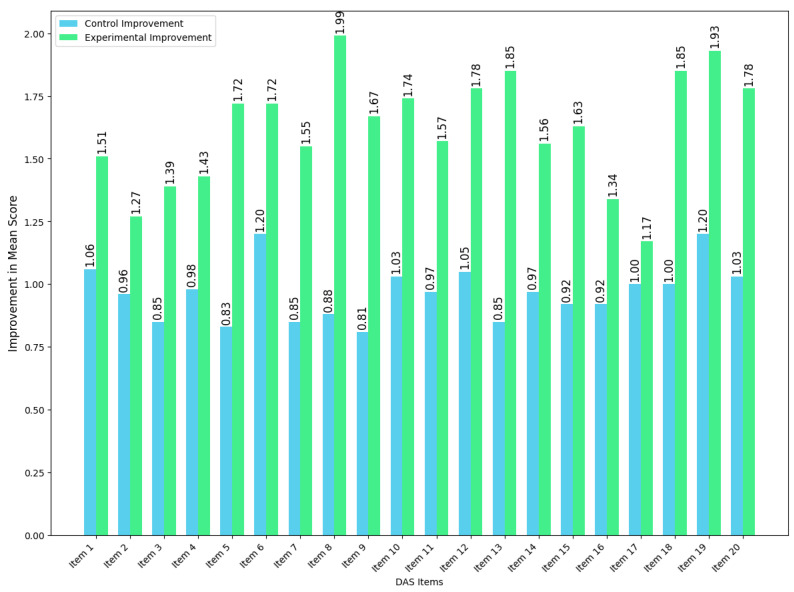
Pre–post improvement in the DAS mean scores.

**Figure 11 bioengineering-12-00987-f011:**
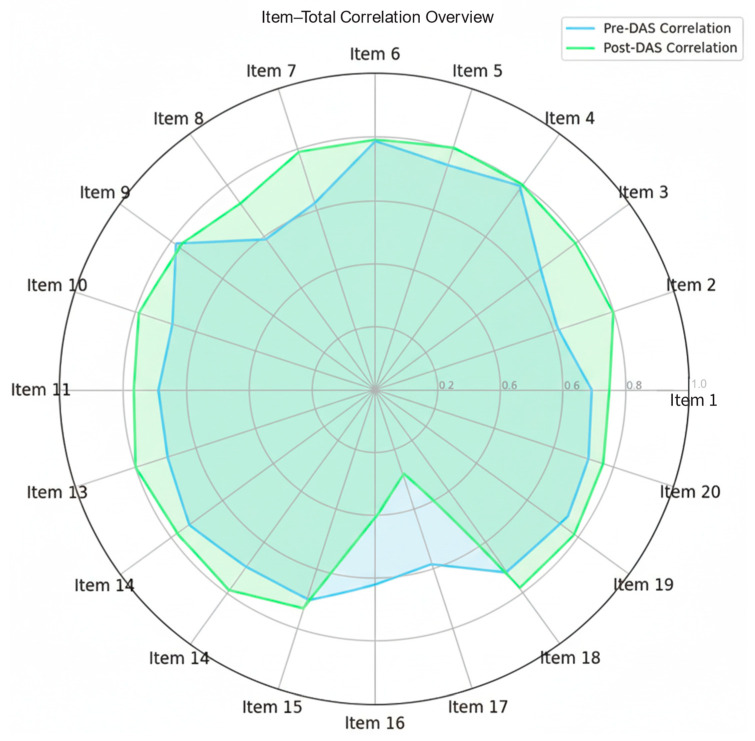
Items correlations for each DAS item, measured before (pre-intervention) and after (post-intervention).

**Table 1 bioengineering-12-00987-t001:** Functions and goals of the components.

Component	Function	Goal
Persona creation	Developing user profiles reflecting real dementia experiences	Personalization, inclusivity, and realism
Narrative structure	Storylines and dialogue trees based on personas and context	Emotional engagement, empathy, and understanding
Interactive mechanics	Decision points and user driven actions in the VR experience	Agency, immersion and experiential learning

**Table 2 bioengineering-12-00987-t002:** Summary of the VR touchpoints in the VDT.

Touchpoint	Description	Dementia Symptom/Behavior	Category	VR Implementation (Unity)
Social Activities	Users accept/reject an invitation	Social withdrawal	Symptom	Button triggers branch to new scene
Familiar Tasks	Attempt to explain Sudoku, but struggle	Cognitive impairment	Symptom	Timer, feedback animation, forced error dialogue
Social Support	Accept/reject help from a passerby	Acceptance of assistance	Coping behavior	NPC animation and dialogue change via user choice
Disorientation of Time	Choose activity at midnight	Time confusion	Symptom	Visual cues, time-of-day setting, branching scene
Emotional Changes	Search for lost item calmly or frantically	Emotional lability	Symptom	Button triggers facial animation, tone shift
Memory Impairment	Explain lost item as misplaced or stolen	Memory distortion	Symptom	Dialogue path selection, object interaction
Judgment Skills	Choose weather-appropriate clothing	Impaired judgment	Symptom	Choice triggers outfit and feedback animation
Family Support	Interact with family member, choose attitude	Social support, family impact	Coping behavior	NPC dialogue/emotion changes with branch selection

Symptom = core dementia symptom; coping/behavioral response = adaptive or relational behavior in response to dementia challenges.

**Table 3 bioengineering-12-00987-t003:** Purposes and tasks of the six phases of VDT development.

Phase	Description
Phase 1: Scenario Recording and Preparation	Footage for each scenario was recorded, with each persona’s home challenges being captured. Emotional changes and daily obstacles were documented. All videos were converted into panoramic MP4 files for VR compatibility.
Phase 2: Video Integration in VR Platform	The edited panoramic videos were imported into Unity (version 2021.3.9f1), which was selected for its robust VR development capabilities and support for interactive features.
Phase 3: Virtual Environment Design	Immersive environments for each persona were designed, with custom skyboxes and realistic materials applied. The render texture was set to 2040 × 1080 pixels to ensure high visual fidelity.
Phase 4. User Interface and Personalization	An interactive VR interface was developed, allowing profiles to be modified, settings to be personalized, and user-specific information to be input, enhancing immersion and ownership.
Phase 5: Interactive Element Integration	Interactive elements, such as decision-making buttons, were embedded into the VR scenes. These allowed user choices to trigger dynamic scenario responses, fostering active engagement.
Phase 6: Deployment and Testing	All scenes were compiled, build settings were configured for Android, and the application was deployed to VR headsets, enabling users to experience the VDT wirelessly and immersively.

**Table 4 bioengineering-12-00987-t004:** Socio-demographic characteristics of the study sample.

Variable	Category	Frequency (n)	Percentage (%)
Age	18–25	72	90.0%
	26–30	5	6.3%
	Below 18	3	3.8%
Gender	Female	48	60.0%
	Male	32	40.0%
Education Level	Bachelor’s degree	61	76.3%
	High school diploma	13	16.3%
	Less than high school diploma	3	3.8%
	Master’s degree or higher	3	3.8%

**Table 5 bioengineering-12-00987-t005:** Descriptive statistics for the pre- and post-intervention DAS scores in both groups.

		Mean	N	Std. Deviation	Std. Error Mean
Pair 1	Leaflet-Pre DAS score	70.72	40	13.493	2.13
	Leaflet-Post DAS score	90.05	40	12.946	2.05
Pair 2	VR-Pre DAS score	74.75	40	13.939	2.20
	VR-Post DAS score	107.23	40	10.847	1.72

**Table 6 bioengineering-12-00987-t006:** Paired samples *t*-test results for changes in the DAS scores.

		95% Confidence Interval of the Difference	t	df	Significance	
					One-Sided *p*	Two-Sided *p*
Pair 1	Leaflet-Pre DAS score -Post DAS score	−15.94	−11.53	39	<0.001	<0.001
Pair 2	VR-Pre DAS score −Post DAS score	−28.268	−15.61	39	<0.001	<0.001

**Table 7 bioengineering-12-00987-t007:** Confidence intervals for the pre- to post-intervention DAS scores.

		Paired Differences			
		Mean	Std. Deviation	Std. Error Mean	95% Confidence Interval of the Difference
					Lower
Pair 1	Leaflet-Pre DAS score—Leaflet-Post DAS score	−19.33	10.60	1.68	−22.72
Pair 2	VR-Pre DAS score—VR-Post DAS score	−32.48	13.16	2.08	−36.68

**Table 8 bioengineering-12-00987-t008:** Paired sample correlations for the pre- and post-intervention DAS scores.

		N	Correlation	Significance	
				One-Sided *p*	Two-Sided *p*
Pair 1	Leaflet-Pre DAS score & Leaflet-Post DAS score	40	0.68	<0.001	<0.001
Pair 2	VR-Pre DAS score & VR-Post DAS score	40	0.46	0.001	0.003

## Data Availability

The data presented in this study are available upon reasonable request from the corresponding author.

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
