# Peer review of "An Innovative Interactive Narrative Persona (INP) Approach for Virtual Reality-Based Dementia Tour Design (VDT) in Rehabilitation Contexts"

_bioengineering, 2025, doi:10.3390/bioengineering12090987_

Round 1
Reviewer 1 Report
Comments and Suggestions for Authors
This manuscript developed and evaluated a Virtual Dementia Tour using an Interactive Narrative Persona framework, demonstrating that immersive VR experiences significantly outperformed traditional leaflet-based education in enhancing public awareness, empathy, and understanding toward people living with dementia.. Below are some questions and suggestions for improvement:
- The title emphasises the application of the Innovative Interactive Narrative Persona method in VR program design. Still, the uniqueness and innovation of this integration are not clearly explained, and the differentiated advantages compared to existing VR design methods are also not clearly articulated.
- The INP and 3I theories involved in this study are more technical implementation frameworks, and the medical and psychological theoretical support related to dementia is obviously insufficient.
- The calculation of the DAS score improvement is inconsistent. The text mentions that the experimental group improved by +32.25, but according to Table 5 data, 107.23 minus 74.75 should be 32.48.
- In the Summary of VR Touchpoints in the VDT, “Acceptance of assistance” and “Family support interaction” are not Dementia Symptoms, but rather coping behaviours or responses.
Author Response
- This manuscript developed and evaluated a Virtual Dementia Tour using an Interactive Narrative Persona framework, demonstrating that immersive VR experiences significantly outperformed traditional leaflet-based education in enhancing public awareness, empathy, and understanding toward people living with dementia. Below are some questions and suggestions for improvement:
- The title emphasises the application of the Innovative Interactive Narrative Persona method in VR program design. Still, the uniqueness and innovation of this integration are not clearly explained, and the differentiated advantages compared to existing VR design methods are also not clearly articulated.
Our response: We thank the reviewer for the constructive feedback. We agree that the uniqueness and novelty of our INP approach should be explicitly discussed in the introduction section. To this end, we have substantially revised the introduction chapter to clarify and articulate the uniqueness and innovation of integrating the INP method within VR program design, emphasizing how it addresses limitations of existing VR approaches (Lines 91-107). We have also added a paragraph to highlight the differential advantages of our approach by contrasting it with conventional VR dementia education (Lines 109-120).
- The INP and 3I theories involved in this study are more technical implementation frameworks, and the medical and psychological theoretical support related to dementia is obviously insufficient.
Our response: Thank you for your valuable feedback. We recognize the importance of grounding dementia care interventions in robust medical and psychological theory. To address this, we have incorporated insights from two of well-established theoretical frameworks in dementia care: Kitwood’s Theory of Person-Centred Care1 and The Biopsychosocial-Ecological Framework 2. Kitwood’s model emphasizes the importance of personhood, recognition, respect, and trust in supporting the wellbeing of people with dementia. On the other hand, the Biopsychosocial-Ecological Framework integrates medical, psychological, and social dimensions—including family dynamics and caregiving networks. By integrating Kitwood’s psychological theory and Podgorski et al.’s holistic medical and ecological perspective, our study is grounded in comprehensive, up-to-date theoretical models from the field of dementia care. We have highlighted this in section 2.2 (Lines 165-192) of the revised manuscript.
- Terkelsen, A. S., Petersen, J. V. & Kristensen, H. K. Mapping empirical experiences of Tom Kitwood’s framework of person-centred care for persons with dementia in institutional settings. A scoping review. Scandinavian Journal of Caring Sciences 34, 6–22 (2020).
- Podgorski, C. A., Anderson, S. D. & Parmar, J. A Biopsychosocial-Ecological Framework for Family-Framed Dementia Care. Front. Psychiatry 12, (2021).
- The calculation of the DAS score improvement is inconsistent. The text mentions that the experimental group improved by +32.25, but according to Table 5 data, 107.23 minus 74.75 should be 32.48.
Our response: Thank you for identifying this discrepancy. We apologize for the oversight in the reported DAS score improvement for the experimental group. The correct mean change in DAS score is indeed 32.48, as calculated from the values in Table 5 (107.23 - 74.75) and consistent with the paired difference of -32.48 reported in Table 6. We have revised the text to accurately reflect the value of 32.48. Additionally, we have ensured that all reported values are rounded to two decimal places for consistency.
- In the Summary of VR Touchpoints in the VDT, “Acceptance of assistance” and “Family support interaction” are not Dementia Symptoms, but rather coping behaviours or responses.
Our response: Thank you for your valuable comment. To improve clarity and conceptual accuracy, we have revised Table 2 to distinguish between core symptoms and coping/behavioral responses. We have also added a legend to the table: “Symptom = core dementia symptom; Coping/Behavioral Response = adaptive or relational behavior in response to dementia challenges.” to distinguish between these categories.
Reviewer 2 Report
Comments and Suggestions for Authors
This study is to explore the impact of virtual reality (VR) on enhancing public awareness and empathy toward dementia.
The study topic is timely. The study motivation and purpose are well addressed. The study background and study design are properly presented. The study analysis and results are valid and applicable.
The following are some comments to improve the quality of the study.
- In the abstract, add which method is utilized to analyze the study.- In the abstract, add the data collection and related study time period for this study.
- In line 100, define interactive narrative persona first.
- In Figures 2 and 3 make sure you have permission from the person in question to have their face shown. Too many participants' personal profiles are publicly released if this study is published. Showing their face and name directly is not recommended.
- In table 4, frequency values should be presented.
- In Figure 10. all mean scores should be presented in numerical values. Current format does not provide exact values so that it can not compare to each other.
- For paired samples t-test, provide equal variance test first, then select proper mean comparison test.
- For study implications, if possible, provide theoretical and practical implications.
Author Response
- This study is to explore the impact of virtual reality (VR) on enhancing public awareness and empathy toward dementia. The study topic is timely. The study motivation and purpose are well addressed. The study background and study design are properly presented. The study analysis and results are valid and applicable. The following are some comments to improve the quality of the study.
- In the abstract, add which method is utilized to analyze the study.- In the abstract, add the data collection and related study time period for this study.
Our response: Thank you for your thoughtful feedback and positive assessment of our study. In response to your suggestions, we have revised the abstract to specify the statistical methods utilized for data analysis (paired and independent samples t-tests). These additions enhance the clarity and transparency of our study methodology.
- In line 100, define interactive narrative persona first.
Our response: Thank you for your valuable suggestion. In response, we have revised Section 2.1 (lines 136–145) in the manuscript to provide a clear definition of the interactive narrative persona (INP) at the outset.
- In Figures 2 and 3 make sure you have permission from the person in question to have their face shown. Too many participants' personal profiles are publicly released if this study is published. Showing their face and name directly is not recommended.
Our response: Thank you for raising this important ethical consideration. We fully agree with your concerns regarding participant privacy and data protection. We have replaced all facial images and real names of participants in Figures 2 and 3 with non-identifiable avatars and pseudonyms. No personal or identifying information is displayed in the revised manuscript. This change ensures the privacy and anonymity of all participants.
- In table 4, frequency values should be presented.
Our response: Thank you for your suggestion. We have revised Table 4 to include both the frequency (n) and percentage (%) values for each socio-demographic variable.
- In Figure 10. all mean scores should be presented in numerical values. Current format does not provide exact values so that it can not compare to each other.
Our Response: Thank you for your helpful suggestion. In the revised manuscript, we have revised Figure 10 to include data labels displaying the exact mean values on each bar for greater clarity.
- For paired samples t-test, provide equal variance test first, then select proper mean comparison test.
Our Response: Thank you for the suggestion. In the revised manuscript, we have updated Section 3.4.2 (Lines 437-441) to include Mauchly’s test of sphericity to verify the sphericity assumption for paired differences in DAS scores for both the control and experimental groups. The results confirm that the assumption was met, justifying the use of standard paired samples t-tests. The presentation of results has also been reordered to report the equal variance test first, followed by the t-test outcomes.
- For study implications, if possible, provide theoretical and practical implications.
Our Response: Thank you for your thoughtful feedback. In response, we have expanded the Discussion section to explicitly address both the theoretical and practical implications of our findings. We added section 4.1(Lines 530 -548): “Theoretical Implications: Experiential Learning and Empathy Development” to discuss how the results inform theories of experiential learning and empathy development (theoretical implications). We have also expanded Section 4.2 to discuss the practical advantages and considerations for implementing VDT interventions in real-world educational and community settings.
Reviewer 3 Report
Comments and Suggestions for Authors
The manuscript addresses an important and timely topic the use of Virtual Reality to enhance empathy and awareness toward dementia through an INP framework. The study is original, methodologically sound, and contributes to the growing evidence on VR-based experiential education. The writing is generally clear, and the manuscript is well structured. Results are consistent and relevant, with appropriate statistical analyses. However, several aspects require clarification and improvement to strengthen the paper before publication.
The INP framework is presented as innovative, but the discussion does not fully differentiate it from existing VR-based dementia interventions. Please clarify what is truly novel compared to previous narrative-based VR methods. The link with rehabilitation contexts is mentioned in the title but not sufficiently developed in the discussion. Its relevance for rehabilitation practice (beyond awareness campaigns) should be clarified.
The sample is composed almost entirely of young adults (90% aged 18–25, mainly students). This strongly limits generalizability to the wider population, caregivers, healthcare professionals, and older adults. This limitation should be discussed more critically.
Randomization is reported but not explained. Was allocation computer-generated or manual?The exclusion criteria (no personal/family history of dementia) may have biased the sample toward very low baseline awareness. Please justify this choice.
The superiority of VR over leaflets is demonstrated, but broader implications for public health campaigns are only briefly mentioned. Cost-effectiveness, feasibility, and technological barriers (e.g., VR headset availability) should be critically addressed. The link to rehabilitation remains vague. Please explain how the INP-VDT could be applied in rehabilitation programs—for patients, caregivers, or staff training.
Some sections (e.g., Unity technical details) are overly descriptive and could be streamlined.
References: Include seminal works on VR and empathy beyond dementia (e.g.: University of Cagliari). Ensure full consistency with MDPI guidelines (remove “Available online” notes).
Figure 11 requires a clearer caption.
Limitations: Structure this section more clearly (sample bias, short-term evaluation, self-report measures). A bullet-point style would improve readability.
Add a paragraph in the Introduction explicitly positioning the INP framework against prior VR dementia interventions.
Expand the Discussion on rehabilitation and scalability.
Author Response
- The manuscript addresses an important and timely topic the use of Virtual Reality to enhance empathy and awareness toward dementia through an INP framework. The study is original, methodologically sound, and contributes to the growing evidence on VR-based experiential education. The writing is generally clear, and the manuscript is well structured. Results are consistent and relevant, with appropriate statistical analyses. However, several aspects require clarification and improvement to strengthen the paper before publication.
- The INP framework is presented as innovative, but the discussion does not fully differentiate it from existing VR-based dementia interventions. Please clarify what is truly novel compared to previous narrative-based VR methods. The link with rehabilitation contexts is mentioned in the title but not sufficiently developed in the discussion. Its relevance for rehabilitation practice (beyond awareness campaigns) should be clarified.
Our Response: Thank you for your insightful feedback. In the revised manuscript, we have explicitly differentiated the INP framework from existing VR-based dementia interventions in Section 4.1, clarifying the specific innovations that distinguish our approach such as the use of theory-driven, diverse personas, branching interactive narratives, and authentic real-world data. Additionally, we have expanded the discussion in Section 4.2.3 to address the relevance and potential applications of the VDT and INP framework in rehabilitation contexts, beyond public awareness campaigns.
- The sample is composed almost entirely of young adults (90% aged 18–25, mainly students). This strongly limits generalizability to the wider population, caregivers, healthcare professionals, and older adults. This limitation should be discussed more critically.
Our Response: Thank you for the feedback. We acknowledge that sample diversity is essential for fully evaluating the effectiveness and relevance of the INP framework. In the revised manuscript, we have highlighted the need for future studies involving more diverse and representative populations as a key direction for further research.
- Randomization is reported but not explained. Was allocation computer-generated or manual?The exclusion criteria (no personal/family history of dementia) may have biased the sample toward very low baseline awareness. Please justify this choice.
Our response: In the revised manuscript (Section 3.2), we have clarified that group allocation was conducted using a computer-generated random number sequence to ensure allocation concealment and minimize selection bias. Regarding the exclusion criteria, we excluded participants with a personal or family history of dementia, as well as current or past caregivers, to reduce potential confounding effects related to prior experience and to minimize the risk of emotional distress.
- The superiority of VR over leaflets is demonstrated, but broader implications for public health campaigns are only briefly mentioned. Cost-effectiveness, feasibility, and technological barriers (e.g., VR headset availability) should be critically addressed. The link to rehabilitation remains vague. Please explain how the INP-VDT could be applied in rehabilitation programs—for patients, caregivers, or staff training.
Our Response: Thank you for the insightful feedback. We have addressed the broader implications of VR-based dementia education in Sections 4.2.2 (Feasibility and Societal Impact) and 4.3 (Recommendations and Future Directions). Specifically, we've discussed the accessibility of wireless and user-friendly VR setups, and the potential for integration in various community settings (see Section 4.3). Regarding the application of the INP-VDT in rehabilitation contexts, we have expanded on this in Section 4.2.3 (Relevance for Rehabilitation Practice). Here, we detail how the INP-based VDT can be adapted for professional training, caregiver support, and patient engagement, as well as its potential to foster communication among healthcare providers, social workers, and families. These examples illustrate how the framework can be implemented for patients, caregivers, and staff training within rehabilitation programs.
- Some sections (e.g., Unity technical details) are overly descriptive and could be streamlined.
Our response: Thank you for the suggestion. We enhanced descriptive sections to improve clarity and maintain focus on the study’s main findings and implications.
- References: Include seminal works on VR and empathy beyond dementia (e.g.: University of Cagliari). Ensure full consistency with MDPI guidelines (remove “Available online” notes).
Our Response: Thank you for the feedback. We have included more seminal works on VR and empathy in the revision. We have also ensured that all references are fully consistent with MDPI guidelines and have removed all “Available online” notes.
- Figure 11 requires a clearer caption.
Our Response: Thank you for your suggestion. We have revised the a more clearer and descriptive caption to enhance clarity and understanding.
- Limitations: Structure this section more clearly (sample bias, short-term evaluation, self-report measures). A bullet-point style would improve readability.
Our Response: Thank you for your suggestion. We have added a Sudy limitations section (Section 4.2.4) to highlight the main limitations to enhance clarity and readability of the manuscript.
- Add a paragraph in the Introduction explicitly positioning the INP framework against prior VR dementia interventions.
Our Response: Thank you for your suggestion. We have added a dedicated paragraph in the Introduction as follows that explicitly positions the INP framework against prior VR dementia interventions.
“Compared to existing VR design methods, the proposed INP framework offers several distinct advantages:
- It supports mindful design by encouraging stakeholders to consider critical factors when creating environments for people with dementia, resulting in greater authenticity and relatability usingINP rather than generic avatars[28].
- It improves social and relational engagement by fostering environments that support storytelling, memory stimulation, and communication [25,28].
- It allows users to participate actively in narrative experiences tailored to diverse dementia profiles [27].
- It offers broader applicability for the general public by addressing key gaps in community-based dementia awareness.
“
This section highlights the unique contributions and advantages of the INP approach compared to existing VR methods.
- Expand the Discussion on rehabilitation and scalability.
Our Response: Thank you for the suggestion. These aspects are now discussed in detail in Sections 4.2.2 (Feasibility and Societal Impact) and 4.2.3 (Relevance for Rehabilitation Practice) of the revised manuscript.
Round 2
Reviewer 1 Report
Comments and Suggestions for Authors
The revision has addressed the comments and met the publication standards.
Reviewer 3 Report
Comments and Suggestions for Authors
The authors have carefully addressed all reviewer comments. The revisions are clear, appropriate, and substantially improve the manuscript. Minor issues raised have been satisfactorily resolved. In my opinion, the responses are adequate, and the revised manuscript is suitable for publication in its current form.